# Early Spinal Injury Stabilization in Multiple-Injured Patients: Do All Patients Benefit?

**DOI:** 10.3390/jcm9061760

**Published:** 2020-06-05

**Authors:** Philipp Kobbe, Patrick Krug, Hagen Andruszkow, Miguel Pishnamaz, Martijn Hofman, Klemens Horst, Carolin Meyer, Max Joseph Scheyerer, Christoph Faymonville, Gregor Stein, Frank Hildebrand, Christian Herren

**Affiliations:** 1Department for Trauma and Reconstructive Surgery, RWTH Aachen University Hospital, Pauwelsstraße 30, 52074 Aachen, Germany; handruszkow@ukaachen.de (H.A.); mpishnamaz@ukaachen.de (M.P.); mhofman@ukaachen.de (M.H.); khorst@ukaachen.de (K.H.); fhildebrand@ukaachen.de (F.H.); cherren@ukaachen.de (C.H.); 2Department of Orthopedics and Trauma Surgery, Rhein-Maas Klinikum GmbH, Mauerfeldchen 25, 52146 Wuerselen, Germany; patrick.krug@gmx.net; 3Department of Medicine, University of Cologne, Joseph-Stelzmann-Str. 24, 50931 Cologne, Germany; carolin.meyer@uk-koeln.de (C.M.); max.scheyerer@uk-koeln.de (M.J.S.); christoph.faymonville@uk-koeln.de (C.F.); 4Department of Orthopedics and Trauma Surgery, Evangelic Hospital Cologne-Weyertal, Weyertal 76, 50931 Cologne, Germany; 5Department of Orthopedics, Trauma and Spine Surgery, Helios Hospital Siegburg, Ringstraße 49, 53721 Siegburg, Germany; gregor.stein@helios-gesundheit.de

**Keywords:** multiple-injured patient, spine injury, AOSpine classification, time to surgery, sepsis

## Abstract

Background: Thoracolumbar spine fractures in multiple-injured patients are a common injury pattern. The appropriate timing for the surgical stabilization of vertebral fractures is still controversial. The purpose of this study was to analyse the impact of the timing of spinal surgery in multiple-injured patients both in general and in respect to spinal injury severity. Methods: A retrospective analysis of multiple-injured patients with an associated spinal trauma within the thoracic or lumbar spine (injury severity score (ISS) >16, age >16 years) was performed from January 2012 to December 2016 in two Level I trauma centres. Demographic data, circumstances of the accident, and ISS, as well as time to spinal surgery were documented. The evaluated outcome parameters were length of stay in the intensive care unit (ICU) (iLOS) and length of stay (LOS) in the hospital, duration of mechanical ventilation, onset of sepsis, and multiple organ dysfunction syndrome (MODS), as well as mortality. Statistical analysis was performed using SPSS. Results: A total of 113 multiple-injured patients with spinal stabilization and a complete dataset were included in the study. Of these, 71 multiple-injured patients (63%) presented with an AOSpine A-type spinal injury, whereas 42 (37%) had an AOSpine B-/C-type spinal injury. Forty-nine multiple-injured patients (43.4%) were surgically treated for their spinal injury within 24 h after trauma, and showed a significantly reduced length of stay in the ICU (7.31 vs. 14.56 days; *p* < 0.001) and hospital stay (23.85 vs. 33.95 days; *p* = 0.048), as well as a significantly reduced prevalence of sepsis compared to those surgically treated later than 24 h (3 vs. 7; *p* = 0.023). These adverse effects were even more pronounced in the case where cutoffs were increased to either 72 h or 96 h. Independent risk factors for a delay in spinal surgery were a higher ISS (*p* = 0.036), a thoracic spine injury (*p* = 0.001), an AOSpine A-type spinal injury (*p* = 0.048), and an intact neurological status (*p* < 0.001). In multiple-injured patients with AOSpine A-type spinal injuries, an increased time to spinal surgery was only an independent risk factor for an increased LOS; however, in multiple-injured patients with B-/C-type spinal injuries, an increased time to spinal surgery was an independent risk factor for increased iLOS, LOS, and the development of sepsis. Conclusion: Our data support the concept of early spinal stabilization in multiple-injured patients with AOSpine B-/C-type injuries, especially of the thoracic spine. However, in multiple-injured patients with AOSpine A-type injuries, the beneficial impact of early spinal stabilization has been overemphasized in former studies, and the benefit should be weighed out against the risk of patients’ deterioration during early spinal stabilization.

## 1. Introduction

The patient with multiple injuries is a unique challenge for the trauma care team, even more so in association with spinal fractures. Traumatic spinal injuries in multiple-injured patients are less common in comparison to extremity injuries but present with an increasing incidence over the last decades [1,2]. Most of these injuries are located within the thoracic and lumbar spine and are accompanied by lung contusion and/or abdominal trauma [3,4]. Beside worse outcomes, a cost analysis showed that apart from injuries of the head, associated spinal lesions in multiple-injured patients result in higher hospitalisation costs and prolonged length of stay (LOS) [5].

Controversy still arises concerning the optimal timing for spinal surgery in the multiple-injured patient. In the literature, the discussion has ultimately focused on the question of early versus late surgical treatment and its consequences on mortality and the patients’ outcome [6,7]. Recent register studies generally advocate that early spinal stabilization is beneficial for multiple-injured patients, mainly due to the increased possibility of early patient mobilisation in the ICU [8]; however, these register studies do not differentiate the wide variety of different spinal injuries. For example, although AOSpine A-type fractures are managed surgically in some countries to maintain sagittal spinal balance, they are—from a biomechanical viewpoint—stable enough to tolerate early mobilisation in the ICU without the risk of neurological compromise. Therefore, a general recommendation to perform early spinal stabilization, which is definitely a surgical burden and second hit in this scenario, may not be justified, and may even jeopardize multiple-injured patients with biomechanically stable spinal injuries.

For this reason, the aim of the present study is to analyse the impact of the timing of spinal surgery in multiple-injured patients both in general and in respect to spinal injury severity.

## 2. Methods

This retrospective two-centre study was conducted on all multiple-injured patients with spinal injuries who were admitted to one of the two level I trauma hospitals between January 2012 and December 2016. Patients were eligible for further analysis if they met the following inclusion criteria: age >16 years; injury severity score (ISS) >16; associated spinal injury within the thoracic or lumbar spine; and a fully available computed tomography scan for spinal injury classification, according to the established AOSpine Classification [9]. Demographic data, circumstances of the accident, overall injury severity score (ISS), and time to spinal surgery (24, 72, and 96 h) were documented. Measured outcome parameters were duration of mechanical ventilation, intensive care unit (ICU) length of stay (iLOS), length of hospital stay (LOS), development of sepsis or multiple organ dysfunction syndrome (MODS), and death. Organ failure was defined according to the sequential organ failure assessment (SOFA) score: a value in one or more organ systems of ≥3 points was defined as single or multiple organ failure. Sepsis was defined as systemic inflammatory response syndrome (SIRS) combined with bacteraemia.

Data were collected from the inhouse clinical documentation system. Counts and frequencies were used to describe the sample. Correlation analysis (Spearman) was used to determine any dependence between the variables. For evaluation of influencing factors on the timing of surgery, a chi-square test and a one-way ANOVA was performed. Fisher’s exact test was used for categorical variables. Multivariable logistic regression analysis was performed using time to surgery, LOS, iLOS, and sepsis as dependent variables. Results are presented as odds ratios (ORs) with 95% confidence intervals. Significance was set at *p* < 0.05 for all statistical tests. A power analysis was carried out with G * Power (3.1.9.3, IBM) and resulted in the following statistical power for each statistical test: 0.998 for the ANOVA, 0.963 for the *t*-test, and 0.999 for both the correlation analysis and the regression analysis.

All testing procedures were performed exploratively, so no adjustment for multiple testing has been made. Statistical analyses were performed with IBM SPSS software (version 23; IBM Corp., Armonk, New York, NY, USA). The study complies with the principles of the Declaration of Helsinki (2013) and was approved by the local ethical committee (#EK056/17).

## 3. Results

### 3.1. Demographics

A total of 250 multiple-injured patients with vertebral fractures within the thoracolumbar spine were identified (Table 1). The population consisted of 72 (29%) female and 178 male (71%) patients with a mean age of 46 ± 19 years. The mean ISS was 24.8 ± 12.6 points. The most common injury pattern was a fall of >3 m (*n* = 102, 41%), followed by car accident (*n* = 39, 16%) and motorcycle accident (*n* = 38, 15%). In 153 patients (61%) the lumbar spine and in 97 patients (38%) the thoracic spine was involved. In 127 patients (51%) spinal stabilization was performed, of whom 14 patients had to be excluded due to missing data (Figure 1). Thus, 113 patients were included for further statistical analysis.

### 3.2. Spinal Injury Severity According to the AOSpine Classification

Within the group of the operatively treated patients, 71 fractures (63%) were classified as AOSpine A-type injuries, whereas 42 fractures (37%) were classified as AOSpine B-/C-type injuries (Figure 2). Seventeen thoracic and 18 lumbar spinal injuries were associated with neurological impairment.

### 3.3. Time to Spinal Surgery

Time to spinal surgery for the 113 multiple-injured patients included in the study is presented in Figure 3. Setting a cutoff of 24 h for spinal surgery, 49 patients (43.4%) were treated within 24 h after trauma. Twenty-one of these patients presented with an AOSpine B-/C-type injury. Increasing the cutoff to 72 h for spinal surgery, 75 patients (66.4%) were treated within 72 h after trauma, of whom 32 patients presented with an AOSpine B-/C-type injury. Further increasing the cutoff to 96 h, 81 patients (71.7%) were treated within 96 h after trauma, of whom 33 patients presented with an AOSpine B-/C-type injury. Regression analysis identified a higher ISS, the thoracic spine injury location, A-type injuries, and the absence of an evident neurological deficit as independent risk factors for a delay in spinal surgery (Table 2).

### 3.4. Impact of Time to Spinal Surgery on Patients´ Outcome

Comparison of multiple-injured patients surgically treated for their spinal injury within 24 h as compared to those surgically treated later than 24 h is shown in Table 3. Multiple-injured patients with their spinal injuries surgically treated within 24 h showed a significantly reduced length of ICU stay by 7 days (7.31 days vs. 14.56 days; *p* < 0.001) as compared to those were operated on later than 24 h while having a comparable overall injury severity (ISS: 23.69 vs. 24.80; *p* = 0.672). Furthermore, the length of hospital stay was significantly reduced by 10 days (23.85 days vs. 33.95 days; *p* = 0.048). The prevalence of sepsis was significantly higher in multiple-injured patients surgically treated for their spinal injury later than 24 h (9 vs. 1, respectively; *p* = 0.023). Mechanical ventilation time was higher in patients surgically treated for their spinal injury later than 24 h, although this did not reach statistical significance (150.94 h vs. 184.05 h, respectively; *p* = 0.525). The prevalence of MODS or death showed no statistical difference between both groups.

Comparison of multiple-injured patients surgically treated for their spinal injury within 72 h as compared to those surgically treated later than 72 h is shown in Table 4. Multiple-injured patients with their spinal injuries surgically addressed later than 72 h showed a significant and twofold longer stay in the ICU (17.5 days vs. 8.46 days, respectively; *p* < 0.001) and in the hospital (44.34 days vs. 22.49 days, respectively; *p* < 0.001). Mechanical ventilation time was higher in patients surgically treated for their spinal injury later than 72 h, although this did not reach statistical significance (229.94 h vs. 141.56 h, respectively; *p* = 0.114). However, patients surgically treated for their spinal injury later than 72 h had a significantly higher ISS as compared to those patients treated within 72 h (28.05 vs. 22.49, respectively; *p* = 0.03). The prevalence of sepsis, MODS, or death showed no statistical difference between both groups.

Comparison of multiple-injured patients surgically treated for their spinal injury within 96 h as compared to those surgically treated later than 96 h is shown in Table 5. Multiple-injured patients for whom surgical treatment of their spinal injuries was postponed for more than 96 h showed a significant and twofold longer stay in the ICU (8.55 days vs. 18.41 days respectively; *p* < 0.001) and in the hospital (23.01 days vs. 45.84 days, respectively; *p* < 0.001). Although the overall injury severity was higher in the group surgically treated for their spinal injuries after 96 h, it did not reach statistical significance (27.72 vs. 22.94; *p* = 0.075). Furthermore, mechanical ventilation time was significantly and almost twofold increased from 137.24 h to 250.81 h (*p* = 0.046). The prevalence of sepsis was also significantly higher in the group surgically treated for their spinal injuries later than 96 h (7 vs. 3, respectively; *p* < 0.005). However, the prevalence of MODS or death was not higher in the group of multiple-injured patients treated for their spinal injuries later than 96 h.

Correlation analysis revealed that there was a significant correlation between time to spinal surgery and outcome parameters (Table 6). Lengths of ICU and hospital stays showed a significant correlation with increasing the time to spinal surgery. The development of sepsis further showed a significant correlation with increasing time to spinal surgery. However, there was no correlation between an increased time to spinal surgery and the development of MODS or death. Regression analysis revealed that increased time to spinal surgery is an independent risk factor for an increased ICU and hospital stay, and spinal stabilization later than 72 h an independent risk factor for the development of sepsis (Table 7).

Subgroup analysis showed that the adverse effect of delayed spinal stabilization is mainly attributable to multiple-injured patients with AOSpine B-/C-type injuries. Regression analysis revealed that in patients with AOSpine A-type spinal injuries, an increased time to spinal surgery was only an independent risk factor for an increased LOS; however, in multiple-injured patients with B-/C-type spinal injuries, an increased time to spinal surgery is an independent risk factor for increased iLOS, LOS, and the development of sepsis (Table 8).

### 3.5. Impact of Spinal Injury Severity on Patients’ Outcome

Comparison of multiple-injured patients with AOSpine A-type injuries compared to those multiple-injured patients with AOSpine B-/C-type injuries is shown in Table 9.

Multiple-injured patients with A-type spinal injuries showed a trend towards a lower overall injury severity compared to multiple-injured patients with B-/C-type spinal injuries (ISS: 22.75 vs. 26.36, respectively; *p* = 0.138). Length of ICU stay was on average 4 days shorter, and time of mechanical ventilation roughly 90 h shorter in multiple-injured patients with A-type spinal injuries compared to those with B-/C-type spinal injuries; however, neither observations reached statistical significance (iLOS: 10.05 days vs. 14.60 days, respectively; *p* = 0.076; mechanical ventilation: 137.69 h vs. 227.85 h, respectively; *p* = 0.104). Time to spinal surgery was non-significantly prolonged in multiple-injured patients with A-type spinal injuries as compared to those multiple-injured patients with B-/C-type injuries (113.33 h vs. 70.90 h, respectively; *p* = 0.206). Multiple-injured patients with B-/C-type spinal injuries, compared to multiple-injured patients with A-type spinal injuries, showed a non-significant, higher prevalence of MODS (4 vs. 2, respectively; *p* = 0.076), sepsis (6 vs. 4, respectively; *p* = 0.071), and death (2 vs. 1, respectively; *p* = 0.310).

Correlation analysis showed a significant correlation between higher spinal injury severity and thoracic spinal injury location; however, there was no correlation between spinal injury severity and any measured outcome parameters, and consequently, regression analysis identified that spinal injury severity was not an independent risk factor for adverse outcome.

### 3.6. Impact of Spinal Injury Location on Patients’ Outcomes

A comparison of multiple-injured patients with thoracic spinal injury location compared to those with lumbar spinal injury location is shown in Table 10.

Multiple-injured patients with a thoracic spine injury showed a similar overall injury severity compared to multiple-injured patients with a lumbar spine injury (ISS: 25.04 vs. 24.19, respectively; *p* = 0.728). Length of ICU stay was significantly increased by an average of 5 days in multiple-injured patients with a thoracic spine injury compared to multiple-injured patients with a lumbar spine injury (14.58 days vs. 9.48 days, respectively; *p* = 0.032), whereas length of hospital stay showed no significant difference—although it was shorter in multiple-injured patients with a thoracic spine injury (25.92 days vs. 32.77 days, respectively; *p* = 0.180). Mechanical ventilation time was increased by an average of almost 90 h in multiple-injured patients with a thoracic spine injury compared to multiple-injured patients with a lumbar spine injury; however, this did not reach statistical significance (223.41 h vs. 136.92 h; *p* = 0.103). Time to spinal surgery was prolonged in multiple-injured patients with thoracic spine injury compared to multiple-injured patients with a lumbar spine surgery (119.24 h vs. 74.24 h; *p* = 0.152). The prevalence of MODS and death was comparable in both groups, whereas the prevalence of sepsis was significantly higher in multiple-injured patients with a thoracic spine injury compared to multiple-injured patients with a lumbar spine injury (8 vs. 2; *p* = 0.011).

Correlation analysis revealed that spinal injury location significantly correlated with patients’ age, the prevalence of sepsis, the time to spinal surgery, and the spinal injury severity. Regression analysis identified spinal injury location to be an independent risk factor for an increased length of ICU stay (*p* = 0.04, Table 7).

## 4. Discussion

The clinical course for multiple-injured patients is determined by the initial trauma (first hit), the initial surgical burden (second hit), and the resulting systemic inflammatory response [10]. In contrast to the established damage control orthopedics (DCO) concept for long-bone fractures, which recommends performing a definite long-bone fracture stabilization later in the course of treatment, early definite spinal stabilization in multiple-injured patients is gaining increasing popularity in recent years [11,12]. This “spine damage control” concept advocates immediate posterior spinal stabilization in multiple-injured patients, with delayed anterior spinal stabilization, if required, later in the course [11,13]. Support of this concept is provided by several studies showing that early spinal stabilization in multiple-injured patients appears to be associated with a beneficial medical and socioeconomic outcome [5,8,14,15].

In general, our data support the concept of early posterior spinal stabilization in multiple-injured patients. Our data show that increasing time to spinal surgery in multiple-injured patients is an independent risk factor for an increased ICU and hospital stay, and that spinal stabilization later than 72 h is an independent risk factor for the development of sepsis. We consider immediate spinal stabilization, as advocated above, as spinal surgery within 24 h after trauma. Applying a 24 h cutoff to our patient population, we are able to show that multiple-injured patients with their spinal stabilization later than 24 h had, on average, a significantly longer stay of 7 days in the ICU, and a significantly longer length of hospital stay of 10 days, while having a comparable overall injury severity, as defined by the ISS, compared to those multiple-injured patients with spinal surgery within 24 h. Furthermore, the duration of mechanical ventilation was increased by approximately 30 h, and the prevalence of sepsis was significantly increased in the case of spinal stabilization later than 24 h. These adverse effects were even more pronounced for 72 h or 96 h as time cutoffs for spinal surgery. Several studies have attributed the beneficial effect of early spinal stabilization mainly to a better patient handling in the ICU, which allows, for example, for prone positioning for the improvement of pulmonary function, or even earlier in- or out-of-bed mobilization [8]. Most of these studies, however, do not consider spinal injury severity from a biomechanical point of view. Thus, the above-mentioned reasons for a beneficial effect of early spinal stabilization may only be true for instable spinal injuries (AOSpine B-/C-type), because stable spinal injuries (AOSpine A-type) may be considered for early mobilization anyway.

Therefore, the “spine damage control” concept has to be interpreted with caution and should not result in the recommendation that in multiple-injured patients, every spinal injury requiring surgery should be stabilized as early as possible at almost any cost. Further subgroup analyzation of our patient population supports a more individualized treatment concept, taking into account the degree of spinal injury stability, as described with the AOSpine classification. In multiple-injured patients with AOSpine B-/C-type spinal injuries, which from a biomechanical view are unstable spinal injuries, stabilization as early as possible appears to be beneficial. In these patients, prolonged time to spinal surgery is an independent risk factor for increased length of stay in the ICU and in hospital, as well as for the development of sepsis. In contrast to these findings, the benefit of an early spinal stabilization for multiple-injured patients with AOSpine A-type spinal injuries appears to be less pronounced. In these biomechanically stable spinal injuries, a prolonged time to spinal stabilization is only an independent risk factor for an increased length of hospital stay. Although we could not observe an adverse effect of the stabilization of AOSpine A-type spinal injuries in our multiple-injured patients, this observation reduces the pressure on the trauma team to deal with AOSpine A-type spinal injuries in the acute phase. Since AOSpine A-type injuries are the predominant spinal injury pattern in our multiple-injured patient population, this may have an important impact on trauma care in the future. Although a differentiation of the spinal injury, according to the AOSpine classification, is important for decisions with regard to surgical timing, the injury type in and of itself is not an independent risk factor for adverse outcomes.

Thoracic spinal injury location has been identified by other studies to adversely influence patients’ outcome [16,17]. Our data reveals that thoracic spinal injury location is an independent risk factor for prolonged ICU stay. This may be explained by two considerations. First, thoracic spinal injury location in our population is associated with delayed spinal stabilization, probably due to a higher grade of required surgical expertise. Secondly, thoracic spinal injuries are probably associated with a higher rate of thoracic injury (for example, lung contusions). In our population, multiple-injured patients with thoracic spinal injuries on average required 90 h more of mechanical ventilation than those multiple-injured patients with lumbar spinal injuries.

Our data show that in our patient population, almost half of the patients (47.8%) were surgically treated for their spinal injury within 24 h, and almost ¾ of the patients (71.7%) within 72 h. Within 24 h, 50% (21/42 patients) of the total AOSpine B-/C-type spinal injuries and 39% (28/71) of the total AOSpine A-type spinal injuries were surgically addressed; thus, the surgeons already recognized the higher surgical urgency of AOSpine B-/C-type injuries. Reasons for a delayed spinal stabilization appear to be multifactorial. Regression analysis identified, in particular, a higher ISS, thoracic spine injury location, A-type injuries, and the absence of an evident neurological deficit as independent risk factors for a delay in spinal surgery.

In our population, 31% of the multiple-injured patients (*n* = 35) presented with neurological impairment due to their spinal trauma. In the literature, urgent surgical spinal treatment is widely accepted in the event of any neurological impairment focusing on isolated spine trauma. The main target is the decompression of the affected nerve structures and the restoration of the correct spinal alignment. The analysis of the STASCI (Surgical Timing in Acute Spinal Cord Injury) study showed that a general improvement of at least two ASIA (American Spinal Injury Association) grades is observable after early decompression within 24 h [18,19]. Our data show that even in multiple-injured patients with different priority surgeries, neurological impairment resulted in an earlier spinal surgery; however, we were not able to assess whether this had a beneficial impact on neurological outcome.

There are some limitations of our study, which merit further comments. Endpoint selection may not be meticulous enough to identify further differences between the treatment groups, and no in-depth information concerning patients’ comorbidities were available for analysis. The clinical course and outcome of multiple-injured patients are influenced by several factors, which might not all be included in our analysis. In particular, spinal trauma with neurological compromise might have a severe impact on outcome parameters of these multiple-injured patients, which might not be valued enough in our data due to early out-of-hospital transfer to neurological rehabilitation units.

Furthermore, the population size is limited as compared to register studies. However, this is the first study to consider spinal injury severity, according to the AOSpine classification, and to correlate this to the impact of surgical timing of spinal injuries. There are several existing fracture classification systems to evaluate the fracture pattern of spinal fractures. Advantages of the AOSpine fracture classification are better reliability and feasibility in clinical practice, in comparison to the thoracolumbar injury classification and severity score (TLICS). Several studies confirmed that the AOSpine classification is superior to the TLICS with regard to reliability for the identification of fracture morphology [20,21].

In conclusion, our data support the concept of early spinal stabilization within 24 h in multiple-injured patients with AOSpine B-/C-type injuries, especially of the thoracic spine. However, in multiple-injured patients with AOSpine A-type injuries, the beneficial impact of early spinal stabilization has been overemphasized in former studies, and the benefit should be weighed against the risk of patients’ deterioration, due to the unneglectable surgical burden of spinal surgery.

## Figures and Tables

**Figure 1 jcm-09-01760-f001:**
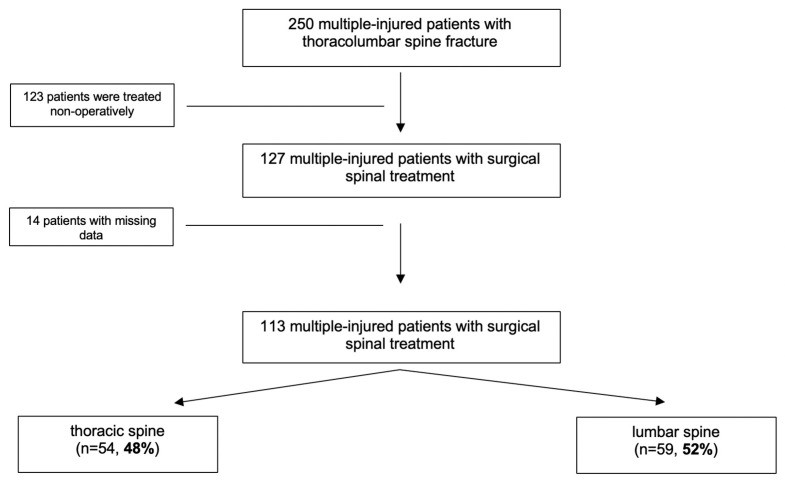
Flowsheet of the included population.

**Figure 2 jcm-09-01760-f002:**
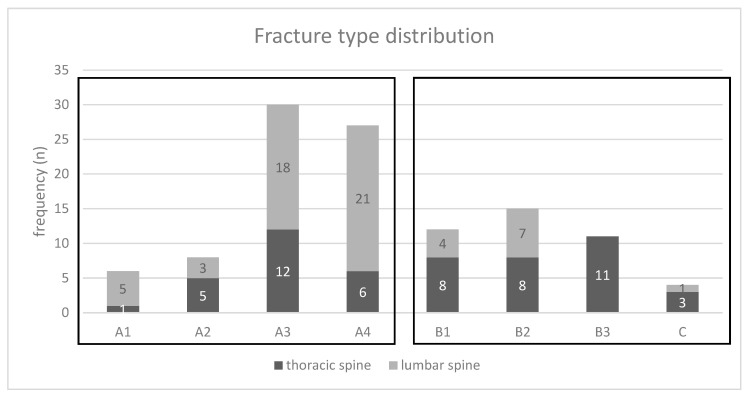
Distribution of the fracture type within the thoracic and lumbar spine of the surgically treated patients.

**Figure 3 jcm-09-01760-f003:**
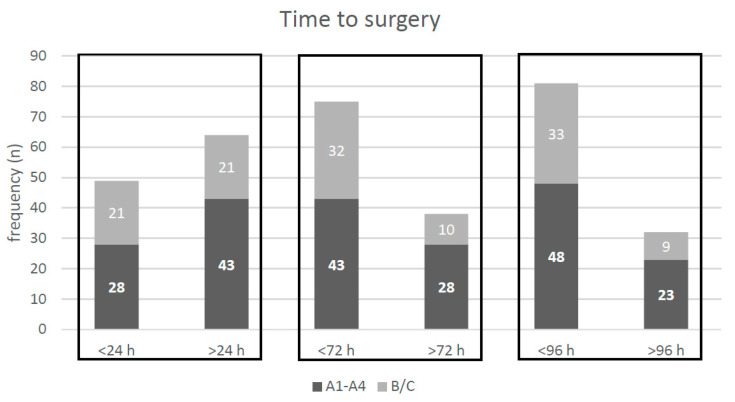
Distribution of the time to surgery related to three defined time points: 24, 72, and 96 h.

**Table 1 jcm-09-01760-t001:** Baseline demographics for spinal fractures in multiple-injured patients.

Demographics	
mean age (SD, years)	46.4 (18.9)
female (%)	72 (28.8%)
male (%)	178 (71.2%)
mean ISS (SD)	24.8 (12.6)
**Mechanism of Injury (*n*)**	
high fall (>3 m)	102
car accident	39
motorcycle accident	38
low fall (<3 m)	26
other	19
bicycle accident	15
pedestrian	11
**AIS ≥3 (*n*)**	
Head and Face	84
Thorax	124
Abdomen	35
Pelvis	54
Extremities	139
**Spine Level (*n*)**	
thoracic	97
lumbar	153

ISS: injury severity score; AIS: abbreviated injury scale.

**Table 2 jcm-09-01760-t002:** Regression analysis for time to surgery (dependent variable: >24 h).

	Regression Coefficient	*p*-Value	OR	95% CI
Age	−0.007	0.604	0.993	0.97–1.02
ISS	0.041	**0.036**	1.042	1.00–1.08
Thoracic spine	1.889	**0.001**	0.151	0.05–0.46
A-type injuries	1.129	**0.048**	3.094	1.01–9.48
No neurological deficit	2.205	**<0.001**	9.069	2.82–29.20

Bold marked values indicate significant parameters. OR: odds ratio.

**Table 3 jcm-09-01760-t003:** Stratification analysis (ANOVA and chi-square test) for the time to surgery (<24 vs. >24 h).

Parameter		≤24 h	>24 h	*p*-Value
	Time			
*n*		49	64	
Age (years)		44.07 ± 18.79	47.91 ± 18.40	0.267
ISS		23.69 ± 14.76	24.80 ± 11.29	0.672
LOS (days)		23.85 ± 19.14	33.95 ± 30.63	**0.048**
ICU (days)		7.31 ± 7.19	14.56 ± 13.93	**<0.001**
mechanical ventilation (h)		150.94 ± 266.86	184.05 ± 279.64	0.525
sepsis (*n*)		1	9	**0.023**
MODS (*n*)		2	4	0.690
death (*n*)		2	1	0.592

Significant *p*-values are marked in bold. LOS: length of stay in the hospital; ICU: intensive care unit; MODS: multiple organ dysfunction syndrome.

**Table 4 jcm-09-01760-t004:** Stratification analysis (ANOVA and chi square test) for the time to surgery (<72 vs. >72 h).

Parameter		≤72 h	>72 h	*p*-Value
	Time			
*n*		75	38	
Age (years)		45.70 ± 18.65	47.14 ± 18.70	0.700
ISS		22.49 ± 12.80	28.05 ± 12.60	**0.03**
LOS (days)		22.49 ± 16.99	44.34 ± 35.90	**<0.001**
ICU (days)		8.46 ± 9.13	17.50 ± 14.79	**<0.001**
mechanical ventilation (h)		141.56 ± 258.44	229.94 ± 297.85	0.114
sepsis (*n*)		4	6	0.070
MODS (*n*)		4	2	1.00
death (*n*)		3	0	0.551

Significant *p*-values are marked in bold.

**Table 5 jcm-09-01760-t005:** Stratification analysis (ANOVA and chi-square test) for the time to surgery (<96 vs. >96 h).

Parameter		≤96 h	>96 h	*p*-Value
	Time			
*n*		81	32	
Age (years)		45.15 ± 18.71	48.84 ± 18.31	0.340
ISS		22.94 ± 12.50	27.72 ± 13.68	0.075
LOS (days)		23.01 ± 16.81	45.84 ± 37.83	**<0.001**
ICU (days)		8.55 ± 9.94	18.41 ± 15.43	**<0.001**
Mechanical ventilation (hours)		137.24 ± 252.88	250.81 ± 308.83	**0.046**
Sepsis (*n*)		3	7	**<0.005**
MODS (*n*)		4	2	0.673
Death (*n*)		3	0	0.562

Significant *p*-values are marked in bold.

**Table 6 jcm-09-01760-t006:** Correlation analysis (Spearman) for time to surgery.

Parameter	Time to Surgery
Age	0.31
ISS	0.25
LOS	**<0.001**
ICU	**<0.001**
Mechanical ventilation	0.08
MODS (*n* = 6)	0.49
Sepsis (*n* = 10)	**0.01**
Death (*n* = 3)	0.28

Significant *p*-values are marked in bold.

**Table 7 jcm-09-01760-t007:** Regression analysis with the dependent variables: LOS (days), length of stay in intensive care (iLOS; days), and sepsis.

	*p*-Value	Regression Coefficient	95% CI
**ICU Stay (iLOS)**			
Age	0.878	0.008	−0.10–0.12
ISS	**<0.001**	0.447	0.27–0.62
Localisation	**0.040**	−4.526	−8.85–−0.20
Fracture type	0.330	2.249	−2.31–6.81
Neurological deficit	0.096	−4.197	−9.15–0.76
Time to surgery (h)	**0.020**	0.015	0.02–0.03
**Hospital Stay (LOS)**			
Age	0.876	0.017	−0.20–0.23
ISS	0.094	0.293	−0.05–0.64
Localisation	0.158	6.170	−2.43–14.77
Fracture type	0.639	2.114	−6.81–11.04
Neurological deficit	0.100	7.987	−1.55–17.53
Time to surgery (h)	**<0.001**	0.045	0.02–0.07
**Sepsis**			
	***p*-value**	**OR**	**95% CI**
Age	0.888	0.997	0.96–1.04
ISS	0.575	0.982	0.92–1.05
Localisation	0.058	0.121	0.01–1.07
Fracture type	0.198	2.974	0.57–15.65
Time to surgery (72 h)	**0.046**	5.543	1.03–29.86

Bold marked values indicate significant parameters.

**Table 8 jcm-09-01760-t008:** Regression analysis for A and B-/C-type fractures with the dependent variables: LOS (days), iLOS (days), and sepsis.

A-Type Fractures (*n* = 71)				B/C-Type Fractures (*n* = 42)		
	*p*-Value	Regression Coefficient	95% CI	*p*-Value	Regression Coefficient	95% CI
**ICU Stay (iLOS)**						
Age	0.59	0.041	−0.11–0.19	0.392	0.067	−0.09–0.23
ISS	**0.003**	0.393	0.14–0.64	**0.002**	−0.4	0.16–0.64
Localisation	**0.028**	−5.889	−11.14–−0.64	0.867	−0.602	−7.83–6.63
Neurological deficit	0.862	−0.593	−7.37–6.19	0.472	−2.661	−10.10–4.78
Time to surgery (h)	0.444	0.005	−0.01–0.02	**<0.001**	0.059	0.03–0.09
**Hospital Stay (LOS)**						
Age	0.922	0.016	−0.31–0.34	0.378	0.132	−0.17–0.43
ISS	0.823	0.058	−0.46–0.58	0.069	0.42	−0.04–0.88
Localisation	0.758	1.699	−9.26–12.66	**0.024**	16.891	2.32–31.46
Neurological deficit	0.08	12.155	−1.49–25.80	0.237	8.455	−5.84–22.75
Time to surgery (h)	**0.009**	0.039	0.01–0.07	**0.004**	0.081	0.03–0.14
**Sepsis**						
	***p*-Value**	**Odds Ratio (OR)**	**95% CI**	***p*-Value**	**Odds Ratio (OR)**	**95% CI**
Age	0.362	0.972	0.91–1.03	0.204	1.074	0.96–1.20
ISS	0.501	0.501	0.93–1.20	0.297	0.914	0.77–1.08
Localisation	0.091	0.091	0.01–1.40	0.999	0	0.01–1.00
Time to surgery (h)	0.857	0.999	0.99–1.01	**0.023**	1.017	1.02–1.03

Bold marked values indicate significant parameters.

**Table 9 jcm-09-01760-t009:** Stratification analysis (ANOVA and chi-square test) of the fracture type (A-type vs. B-/C-type fractures).

Parameter		A-Type Fractures	B-/C-Type Fractures	*p*-Value
	Fracture Type			
*n*		71	42	
Age (years)		46.04 ± 17.68	47.24 ± 20.82	0.746
ISS (points)		22.75 ± 10.72	26.36 ± 14.85	0.138
LOS (days)		29.12 ± 25.02	27.11 ± 21.40	0.678
ICU (days)		10.05 ± 10.94	14.60 ± 15.56	0.076
Mechanical ventilation (hours)		137.69 ± 242.36	227.85 ± 325.54	0.104
Time to surgery (hours)		113.33 ± 186.62	70.90 ± 130.33	0.206
Sepsis (*n*)		4	6	0.071
MODS (*n*)		2	4	0.076
Death (*n*)		1	2	0.310

**Table 10 jcm-09-01760-t010:** Stratification analysis (ANOVA and chi-square test) of the fracture localisation.

Parameter		Thoracic Spine	Lumbar Spine	*p*-Value
	Localisation			
*n*		54	59	
Age (years)		49.79 ± 19.24	42.76 ± 17.65	**0.039**
ISS (points)		25.04 ± 11.69	24.19 ± 14.20	0.728
LOS (days)		25.92 ± 18.05	32.77 ± 32.28	0.180
iLOS (days)		14.58 ± 13.86	9.48 ± 11.41	**0.032**
Mechanical ventilation (hours)		223.41 ± 298.96	136.92 ± 264.55	0.103
Time to surgery (hours)		119.24 ± 190.50	74.24 ± 140.52	0.152
Sepsis (*n*)		8	2	**0.011**
MODS (*n*)		3	3	0.385
Death (*n*)		2	1	0.412

Significant *p*-values are marked bold.

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
