# Peer review of "Early Spinal Injury Stabilization in Multiple-Injured Patients: Do All Patients Benefit?"

_jcm, 2020, doi:10.3390/jcm9061760_

Round 1
Reviewer 1 Report
This is an interesting study and well presented. I liked the use of the AOSpine grading scale
Concerns/ Queries
"Measured outcome parameters were duration of mechanical ventilation, ICU length of stay (iLOS), length of hospital stay (LOS), development of sepsis or multiple organ dysfunction syndrome (MODS), and death."
1) While ICU LOS is a useful healthcare economic metric, in this instance, would the postoperative ICU LOS be a more appropriate parameter to measure in order to better gauge and compare groups?
"Regression analysis identified a higher ISS, thoracic spine injury location, A-type injuries as well as the absence of an evident neurological deficit as independent risk factors for a delay in spinal surgery"
2) More severely injured patients (higher ISS) having a prolonged time to surgery and subsequent adverse postoperative course is a significant confounding variable, which greatly limits any conclusions to be drawn on the impact of timing on spinal surgery.
3) More severely injured patients who had prolonged time to spinal theatre may be due to higher priority operations beforehand. Worse postoperative course and increased sepsis rates could be due to these operations / associated injuries- is there any data on source of sepsis ?
4) neurologically injured patients are typically triaged ahead of non-neurologically injured and so this finding is not surprising. However, neurological injured patients have considerably different care needs to non-neurologically injured patients and so both of these cohorts analysed as one represents a significant confounding factor.
5) A reported limitation is that of sample size, especially in comparison to register studies. Was a power analysis performed to gauge required patient numbers? There has been enough studies performed previously investigating the impact of surgical timing of spinal injuries to perform a power analysis.
The authors define 'immediate' stabilisation as within 24 hours, and also use 72 hours and 96 hours as timepoints.
6) Why was the 48 hour timepoint not used?
7) the authors conclude by supporting 'early' stabilisation... What is 'early' as defined by the authors and how does this compare to existing literature?
8) any information on patient baseline comorbidities and whether this may have affected to time to surgery and postoperative outcomes?
Reviewer 2 Report
Interesting study. Carries the usual stigmata of a retrospective study.
Need to define the AO types (Table).
Include at least in the discussion the TLICS score.
Reviewer 3 Report
The authors conducted a retrospective study to assess the impact of surgical timing on spinal trauma in multiple-injured patients. I have a few questions and suggestion:
- Table 1 should include columns showing the two population groups and the demographic distribution.
- No information was provided on patients who sustained a spinal cord injury secondary to the trauma. This could be relevant since we know from previous work that the severity of spinal cord injury and timing of operation can have a significant impact on the development of sepsis, the length of stay, mortality, as well as the overall outcome.
- Is there any information regarding the development of adverse events (apart from sepsis)? Whether delayed surgery could impact the development of adverse events? Which, in turn, could contribute to the outcome and mortality?
- Although less important than the questions above, I wonder if the authors have any information or have looked at the type of procedure (anterior, posterior, combined, staged, etc.) playing a role in impacting the length of hospital or ICU stay, as well as sepsis?
- The authors showed that the time of mechanical ventilation and ICU stay is increased for patients surgically treated beyond 24 hours of the initial injury. It wasn’t clear to me whether the authors measured the time for mechanical ventilation and length of ICU stay from the time of admission or from time of surgery? Given the study's purpose to understand the impact of early surgery, I would imagine the time is measured from the time of surgery to time of extubation postoperatively or discharge from ICU. Since the population in this study are polytrauma patients, and as reported by the authors, increased ISS has been demonstrated to delay surgical intervention. If the time of mechanical ventilation is measured from the time of admission (most likely the severely injured patients are), then the increased ventilation time and ICU stay in patients treated beyond 24 hours of injury could simply be a reflection of the severity of initial trauma and prolonged time required to stabilize and optimize patient for surgery.
- Finally, the literature review needs to be broadened and more extensive. Rather than focusing on spinal trauma, consideration should be given to other major predictors of outcome, length of stay, and survival. Several suggested articles to consider including in the discussion:
- Fehlings, M.G., et al., Early versus delayed decompression for traumatic cervical spinal cord injury: results of the Surgical Timing in Acute Spinal Cord Injury Study (STASCIS). PLoS One, 2012. 7(2): p. e32037.
- Fehlings, M.G., et al., A Clinical Practice Guideline for the Management of Patients With Acute Spinal Cord Injury and Central Cord Syndrome: Recommendations on the Timing (</=24 Hours Versus >24 Hours) of Decompressive Surgery. Global Spine J, 2017. 7(3 Suppl): p. 195S-202S.
- Wilson, J.R., D.W. Cadotte, and M.G. Fehlings, Clinical predictors of neurological outcome, functional status, and survival after traumatic spinal cord injury: a systematic review. J Neurosurg Spine, 2012. 17(1 Suppl): p. 11-26
- Jaja, B.N.R., et al., Association of Pneumonia, Wound Infection, and Sepsis with Clinical Outcomes after Acute Traumatic Spinal Cord Injury. J Neurotrauma, 2019.
- Jiang, F., et al., Acute Adverse Events After Spinal Cord Injury and Their Relationship to Long-term Neurologic and Functional Outcomes: Analysis From the North American Clinical Trials Network for Spinal Cord Injury. Crit Care Med, 2019. 47(11): p. e854-e862.
- Kopp, M.A., et al., Long-term functional outcome in patients with acquired infections after acute spinal cord injury. Neurology, 2017. 88(9): p. 892-900
- Failli, V., et al., Functional neurological recovery after spinal cord injury is impaired in patients with infections. Brain, 2012. 135(Pt 11): p. 3238-50.
